# FairFedMed: Achieving Equity in Medical Federated Learning via FairLoRA

## Abstract

Fairness remains a critical concern in healthcare, where unequal access to services and treatment outcomes can adversely affect patient health. While Federated Learning (FL) presents a collaborative and privacy-preserving approach to model training, ensuring fairness is challenging due to heterogeneous data across institutions, and current research primarily addresses non-medical applications. To fill this gap, we introduce *FairFedMed*, the first FL dataset specifically designed to study group fairness (i.e., demographics) in the medical field. It consists of paired 2D SLO funfus images and 3D OCT B-Scans from 15,165 glaucoma patients, along with six different demographic attributes. Existing state-of-the-art FL models may work well for natural images but often struggle with medical images due to their unique characteristics. Moreover, these models do not sufficiently address performance disparities across diverse demographic groups. To overcome these limitations, we propose *FairLoRA*, a novel fairness-aware FL framework based on singular value decomposition(SVD)-based low-rank approximation. *FairLoRA* incorporates customized singular value matrices for each demographic group and shares singular vector matrices across all demographic groups, ensuring both model equity and computational efficiency. Experimental results on the *FairFedMed* dataset demonstrate that *FairLoRA* not only achieves state-of-the-art performance in medical image classification but also significantly improves fairness across diverse populations. Our code and dataset can be accessible via the Github anonymous link: `https://github.com/Anonymouse4Science/FairFedMed-FairLoRA.git`

## 1 Introduction

Fairness is a critical issue in healthcare, as unequal access to services and disparities in treatment outcomes can adversely affect patient health and quality of life. The increasing reliance on artificial intelligence complicates this challenge, as many models are trained on data from specific populations that do not adequately represent all demographic groups. To ensure equitable healthcare, it is essential for models to achieve strong overall performance while consistently performing well across diverse populations, making fairness a key challenge in medical applications.

A major barrier to achieving fairness in the medical field is the challenge of sharing sensitive patient data across institutions due to intellectual property and privacy concerns. Traditional centralized training methods require aggregating data from multiple sources, which poses significant risks of data breaches and privacy violations. This limitation hinders multi-institution collaborations, where diverse datasets are crucial for developing accurate and fair models. Federated Learning (FL) McMahan et al. (2017); Ezzeldin et al. (2023) has emerged as a promising solution to address this barrier. FL allows institutions to jointly train models in a distributed fashion without sharing raw data, thereby protecting patient data privacy. This decentralized method keeps sensitive information local, reducing privacy risks while enhancing model robustness and generalizability.

While FL addresses privacy concerns, it encounters challenges related to fairness due to varying data distributions across institutions. Each client may have different local data and demographic distributions, making it complex to ensure fairness within the FL framework. Research on fairness in FL is categorized into site fairness and group fairness, as illustrated in Fig. 1. Site fairness ensures the model performs equally across institutions, preventing any site from suffering negative

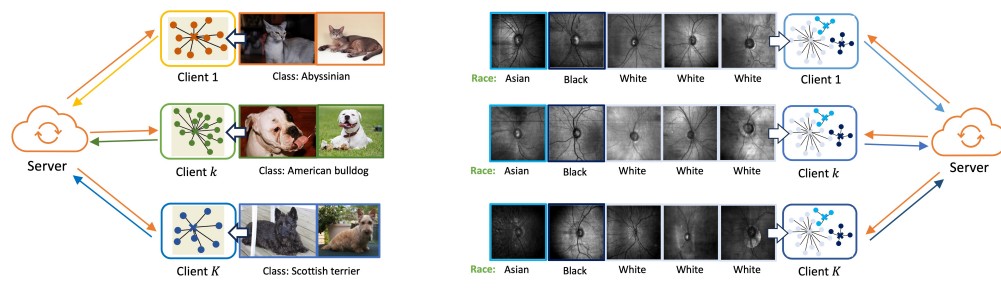

(a) Site fairness                                     (b) Group fairness

Figure 1: Comparison between site fairness and group fairness in federated learning. Site fairness guarantees consistent model accuracy across each client's local dataset, while group fairness ensures equitable performance across different demographic groups within those datasets.

impacts from differences in data quantity or quality. Group fairness ensures that the model performs equitably across various demographic attributes such as different racial groups or ethnic groups. In healthcare, group fairness holds profound significance, as misdiagnosis in certain demographic groups can lead to severe consequences for patients.

Unfortunately, current research on group fairness in FL mainly focuses on non-medical tasks, such as recommendation systems (Agrawal et al., 2024) and financial transactions (Badar et al., 2024). In the medical field, cross-institution demographic differences are common due to geographic heterogeneity (e.g., hospitals in Asia and hospitals in North America), but research on how to manage these differences within the FL framework is relatively limited. Bridging this gap is crucial for developing models that promote health equity and uphold the principle of justice.

In this paper, we address the shortage of medical datasets for fairness-aware FL by introducing the *FairFedMed* dataset for eye disease. This dataset includes 15,165 images from real-world clinics, consisting of paired 2D scanning laser ophthalmoscop (SLO) fundus images and 3D optical coherence tomography (OCT) B-Scans from 15,165 patients. To our knowledge, it is the first FL dataset for group fairness that includes both real-world 2D and 3D medical images, including six different demographic attributes: age, gender, race, ethnicity, preferred language, and marital status. The dataset is partitioned into multiple clients with varied demographic distributions reflecting a real-world FL setting for group fairnress.

Existing state-of-the-art FL methods perform well with natural images but often struggle with medical images due to their unique characteristics. Moreover, these models fail to sufficiently address group fairness across different demographics. To overcome these limitations, we propose *FairLoRA*, a fairness-aware FL framework for eye disease classification. This framework is based on CLIP (Radford et al., 2021) and introduces fairness into the singular value decomposition (SVD) low-rank approximation method. Specifically, to preserve unique intra-group characteristics, FairLoRA customizes singular value matrices for each demographic group while sharing singular vector matrices across all groups to capture inter-group relationships. Local models train a set of low-rank matrices on their local data, and the global model integrates matrices from different clients, allowing the model to aggregate global knowledge about demographic attributes from distributed data. This collaborative approach ensures that the model is not biased toward any particular client or demographic group. Experimental results on the FairFedMed dataset demonstrate that our model achieves state-of-the-art performance in medical image classification while ensuring equitable outcomes across demographic groups. Our main contributions are summarized as follows:

- We introduce FairFedMed, the first FL dataset for group fairness that includes real-world paired 2D and 3D medical images, along with six different demographic attributes.
- We create an experimental setup that divides the dataset into multiple distinct clients with varied demographic distributions, simulating a real-world FL environment with group fairness. We benchmark several state-of-the-art FL methods under this setting.
- We propose FairLoRA, a fairness-aware FL framework for eye disease classification. It customizes singular value matrices for each group to preserve unique characteristics while sharing singular vector matrices across groups to capture inter-group relationships.

## 2 RELATED WORK

**Fairness Learning in Medical Imaging:** Fairness learning in medical imaging aims to reduce biases and ensure equitable outcomes for all patient groups, particularly underrepresented minorities. Currently, most research focuses on fairness models and datasets in classification tasks, with limited attention given to cross-domain fairness. Publicly available datasets, such as CheXpert (Irvin et al., 2019), MIMIC-CXR (Johnson et al., 2019), and Fitzpatrick17k (Groh et al., 2021), support fairness studies but often lack comprehensive identity attributes, focus predominantly on 2D images, and do not include federated clients/sites representing diverse demographic distributions, limiting their applicability to federated learning tasks or 3D medical imaging. Recent methods (Quadrianto et al., 2019; Ramaswamy et al., 2021; Zhang & Sang, 2020; Park et al., 2022; Roh et al., 2020; Sarhan et al., 2020; Zafar et al., 2017; Zhang et al., 2018; Wang et al., 2022; Kim et al., 2019; Lohia et al., 2019) have made progress in algorithmic fairness for medical imaging, but these approaches mainly address bias within individual clients or sites. The performance of multiple clients in a federated learning setup remains largely unexplored, representing a significant gap in fairness research.

**Federated Learning (FL)** is a decentralized machine learning paradigm enabling multiple clients to collaboratively train a global model while keeping their local data private. FL approaches typically fall into two categories: traditional fully parameter-updated models and prompt learning methods. Traditional FL methods (McMahan et al., 2017; Li et al., 2020; 2021; Mendieta et al., 2022) aggregate model parameters from distributed clients to update a global model, allowing collaborative learning while preserving low communication. FedAvg (McMahan et al., 2017) averages local model updates, providing a straightforward and effective solution for various tasks. In contrast, FedProx (Li et al., 2020) adds a proximal term to the objective function, improving stability and performance in heterogeneous environments. Prompt learning methods (Guo et al., 2023; Zhao et al., 2022; Su et al., 2022; Li et al., 2024) customize task-specific text prompts for each client, enabling local and global communication without altering model parameters. PromptFL (Guo et al., 2023) allows clients to train soft prompts instead of the entire model, significantly reducing aggregation overhead and accelerating local training. FedOTP (Li et al., 2024) balances global consensus and local personalization by learning both global and local prompts, using Optimal Transport to align local visual features with prompts and address heterogeneities such as label and feature shifts. The latest methods (Yi et al., 2023; Sun et al., 2024) introduce LoRA (Hu et al., 2022) into FL models to achieve a balance between performance and communication cost.

**Fairness in Federated Learning:** Fairness in FL has gained attention due to its unique challenges compared to centralized learning. Research on fairness in FL can be categorized into two main areas: *Site Fairness* and *Group Fairness*, as illustrated in Fig. 1. *Site Fairness* (Pan et al., 2024b; Cheng et al., 2024; Pan et al., 2024a) ensures equitable handling of model updates from clients, especially when data quality, quantity, or distribution varies. Disparities can lead to models favoring clients with larger or higher-quality datasets. Techniques have been developed to equalize client influence during model aggregation, such as FedLF (Pan et al., 2024b), which uses multi-objective optimization to minimize gradient conflicts and promote equitable model improvements. *Group Fairness* (Ezzeldin et al., 2023; Agrawal et al., 2024; Badar et al., 2024) focuses on equitable model performance across demographic groups within clients' local datasets. FairFed (Ezzeldin et al., 2023) implements a fairness-aware aggregation method, enabling local debiasing and adjusting aggregation weights based on local and global fairness assessments. FairTrade (Badar et al., 2024) employs multi-objective and Bayesian optimization to balance fairness and accuracy. However, most existing studies focus on non-medical applications, like recommendation systems, where biased outcomes are less critical. Despite the urgent need for fairness in healthcare, research on group fairness in federated learning for medical applications is limited, revealing a significant gap in this field.

## 3 FAIRFEDMED DATASET ANALYSIS

To address the shortage of medical datasets for fairness-aware FL, we propose FairFedMed dataset. It contains 15,165 samples from 15,165 subjects with an average age of $61.3 \pm 16.3$ years. Each sample includes both 2D scanning laser ophthalmoscopy (SLO) fundus image and 3D optical coherence tomography (OCT) B-scans, with each containing 128 B-scan images. These images are visualized in the **Appendix A**. Within this dataset, we have six demographic attributes including age, gender, race, ethnicity, preferred language, and marital status. The demographic distributions

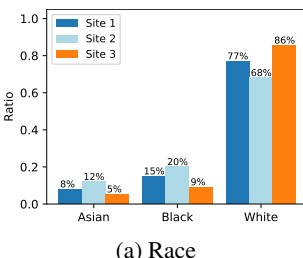
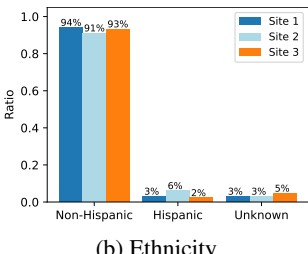
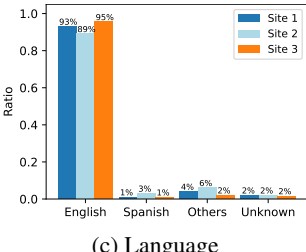

|  (a) Race | (b) Ethnicity | (c) Language |

Figure 2: Demographic distribution across three sites: significant fairness disparities among groups, such as Asian, Black and White in the Race attribute.

are as follows: **Gender:** Female: 57.0%, and Male: 43.0%; **Race:** Asian: 8.4%, Black: 14.7%, and White: 76.9%. **Ethnicity:** Non-Hispanic: 92.7%, Hispanics: 3.7%, Unknown: 3.5%. **Preferred Language:** English: 92.5%, Spanish: 1.6%, Others: 4.0%, and Unknown: 1.9%. **Marital Status:** Married or Partnered: 57.6%, Singe: 26.1%, Divorced: 6.8%, Legally Separated: 0.9%, Windowed: 6.1%, and Unknown: 2.4%. The glaucomatous status of subjects is defined through a comprehensive clinical measurements, where glaucoma and non-glaucoma samples account for 49.0% and 51.0%.

In this work, we divide all subjects into three separate sites, with 80% of the data at each site used for model training and 20% reserved for testing. We focus on three demographic attributes including race, ethnicity and language with varying cross-site distributions. The demographic distributions across three sites are summarized in 2. Given the comprehensive image modalities and demographic attributes of subject samples, our dataset can be used to study different FL settings, *i.e.,* varying number of sites and cross-site image modality differences.

# 4 METHOD

## 4.1 PRELIMINARY

Singular Value Decomposition (SVD) Golub & Reinsch (1971) decomposes a matrix into orthogonal components, providing insights into the geometry of linear transformations and offering a powerful tool for dimensionality reduction. To improve efficiency, the Compact SVD is proposed to reduce the size of the decomposed matrix by retaining only the $r$ largest singular values and their corresponding singular vectors. For a matrix $\boldsymbol{A} \in \mathbb{R}^{m \times n}$, the compact matrix representation can be represented as

$$\boldsymbol{A} \approx \boldsymbol{U}\boldsymbol{S}\boldsymbol{V}^{\top}, \tag{1}$$

where $\boldsymbol{U} \in \mathbb{R}^{m \times r}$ and $\boldsymbol{V} \in \mathbb{R}^{n \times r}$ contain the left and right singular vectors corresponding to the largest $r$ singular values in $\boldsymbol{S} \in \mathbb{R}^{r \times r}$. Here, $r \ll \min(m, n)$ is the rank of $\boldsymbol{A}$.

The SVD-based Adaptation Zhang et al. (2023) extends this concept to model fine-tuning by applying low-rank updates, thereby reducing the number of trainable parameters while preserving model performance, making it especially suitable for optimizing large-scale models. The low-rank adaptation (LoRA) Hu et al. (2022) is essentially the simplest form of SVD, where the singular value matrix $\boldsymbol{S}$ is omitted. In LoRA, the weight matrix is expressed as: $\boldsymbol{W} = \boldsymbol{W}_0 + \Delta \boldsymbol{W} = \boldsymbol{W}_0 + \boldsymbol{U}\boldsymbol{V}^{\top}$, where $\boldsymbol{W}_0$ is the full-rank weight matrix, and $\Delta \boldsymbol{W} = \boldsymbol{U}\boldsymbol{V}^{\top}$ represents the low-rank update. By contrast, the standard SVD-based LoRA includes the singular value matrix $\boldsymbol{S}$, whose updated weight matrix can be formulated as $\boldsymbol{W} = \boldsymbol{W}_0 + \Delta \boldsymbol{W} = \boldsymbol{W}_0 + \boldsymbol{U}\boldsymbol{S}\boldsymbol{V}^{\top}$. The inclusion of the diagonal matrix $\boldsymbol{S}$ enables precise control over the contribution of each singular vector component to the update. Since $\boldsymbol{S}$ contains fewer parameters (just scalars along the diagonal), it offers greater flexibility with minimal parameter overhead.

## 4.2 FEDERATED LEARNING USING LORA

In Federated Learning, the model is trained collaboratively across multiple clients, each with its local data. Let the global model weights be denoted by $\overline{\boldsymbol{W}}^t$ at training round $t$. Each client $k$ maintains a local copy of the model weights, $\boldsymbol{W}_k^t$, and updates its local model based on its own data. The local update for client $k$ is computed as: $\boldsymbol{W}_k^{t+1} = \overline{\boldsymbol{W}}^t - \eta \nabla \mathcal{L}_{D_k}(\overline{\boldsymbol{W}}^t)$, where $\eta$ is the

Figure 3: Overview of fairness-aware federated learning model: FairLoRA, where singular value matrices are customized for each demographic group, such as 'Black', 'Asian', 'White'.

learning rate, and $\nabla \mathcal{L}_{D_k}(\overline{\boldsymbol{W}}^t)$ is the gradient of the loss function with respect to the local data of the $k$ client. After the local updates, the clients communicate their updated weights to the central server, which aggregates them to update the global model: $\overline{\boldsymbol{W}}^{t+1} = \sum_{k=1}^K \alpha_k \boldsymbol{W}_k^{t+1}$, where $\alpha_k = n_k/n$ represents the weights assigned to each client, with $n_k$ denoting the number of data samples at client $k$ and $n = \sum_{k=1}^K n_k$ being the total number of samples across all clients. This weighted average allows clients with larger datasets to contribute proportionally to the update of the global model.

To enhance effective fine-tuning, we adapt a low-rank adaptation within the FL framework, enabling more effective and efficient parameter updates across decentralized datasets. For each client $k$, the model weights incorporating low-rank adaptation at round $t$ can be expressed as: $\boldsymbol{W}_k^t = \boldsymbol{W}_0 + \Delta \boldsymbol{W}_k^t$, where $\boldsymbol{W}_0$ is the pre-trained model weights and $\Delta \boldsymbol{W}_k^t$ is the low-rank update in the client $k$. The low-rank term can be implemented either using LoRA or SVD-based LoRA, such that $\Delta \boldsymbol{W}_k^t = \boldsymbol{U}_k^t \boldsymbol{V}_k^{t\top}$ for LoRA or $\Delta \boldsymbol{W}_k^t = \boldsymbol{U}_k^t \boldsymbol{S}_k^t \boldsymbol{V}_k^{t\top}$ for SVD-based LoRA. The local update for client $k$ depends on the update of low-rank term, which is computed as $\Delta \boldsymbol{W}_k^{t+1} = \Delta \overline{\boldsymbol{W}}^t - \eta \nabla \mathcal{L}_{D_k}(\Delta \overline{\boldsymbol{W}}^t)$. Consequently, the global aggregation on the low-rank term is computed as $\overline{\boldsymbol{W}}^{t+1} = \boldsymbol{W}_0 + \Delta \overline{\boldsymbol{W}}^{t+1} = \boldsymbol{W}_0 + \sum_{k=1}^K \alpha_k \Delta \boldsymbol{W}_k^{t+1}$. This adaptation enables the model to leverage low-rank approximations for the local data at each client, improving computational efficiency while preserving the integrity of the learned representations.

## 4.3 FEDERATED LEARNING USING FAIRLORA

Considering the inherent demographic factors in medical data, the primary challenge is to ensure that the federated model facilitates equitable training across diverse client datasets. To address this issue, we propose FairLoRA, whose overview is shown in Fig. 3. FairLoRA allows for the specification of customized singular value matrices for each demographic group within the low-rank adaptation framework. This personalization enhances the fairness awareness concerning demographic factors. In FairLoRA, the low-rank adaptation for the $k$-th client are represented as follows:

$$\boldsymbol{W}_k^t = \boldsymbol{W}_0 + \Delta \boldsymbol{W}_k^t = \boldsymbol{W}_0 + \boldsymbol{U}_k^t \left(\sum_{g \in \mathcal{G}} \pi_g \boldsymbol{S}_{k,g}^t\right) \boldsymbol{V}_k^{t\top}, \tag{2}$$

where $\Delta \boldsymbol{W}_k^t = \boldsymbol{U}_k^t \left(\sum_{g \in \mathcal{G}} \pi_g \boldsymbol{S}_{k,g}^t\right) \boldsymbol{V}_k^{t\top}$ represents the fairness-aware low-rank update, whose architecture is illustrated in the right of Fig. 3. On one hand, the customized singular value matrix $\boldsymbol{S}_{k,g}^t$ is customized for each demographic group $g$, preserving the unique characteristics. The vector $[\pi_1, \cdots, \pi_g, \cdots, \pi_{|\mathcal{G}|}]$ is a one-hot encoding, ensuring that only the target demographic group has a non-zero value, as $\sum_{g \in \mathcal{G}} \pi_g = 1$. On the other hand, the left and right singular vector matrices $\boldsymbol{U}_k^t$ and $\boldsymbol{V}_k^t$ are shared among all demographic groups, allowing the model to capture inter-group relationships. This design can achieve the trade-off between intra-group and inter-group communication. After the local updates, the global aggregation is computed as follows:

$$\overline{\boldsymbol{W}}^t = \boldsymbol{W}_0 + \Delta \overline{\boldsymbol{W}}^t = \boldsymbol{W}_0 + \overline{\boldsymbol{U}}^t \left(\sum_{g \in \mathcal{G}} \pi_g \overline{\boldsymbol{S}}_g^t\right) \overline{\boldsymbol{V}}^{t\top} \tag{3}$$

where $\overline{\boldsymbol{U}}^t$, $\overline{\boldsymbol{V}}^t$ and $\overline{\boldsymbol{S}}_g^t$ are the shared parameters across all clients to facilitate the communication of global knowledge. Specifically, $\overline{\boldsymbol{U}}^t = \sum_{k=1}^K \alpha_k \boldsymbol{U}_k^t$ and $\overline{\boldsymbol{V}}^t = \sum_{k=1}^K \alpha_k \boldsymbol{V}_k^t$ denote the averaged

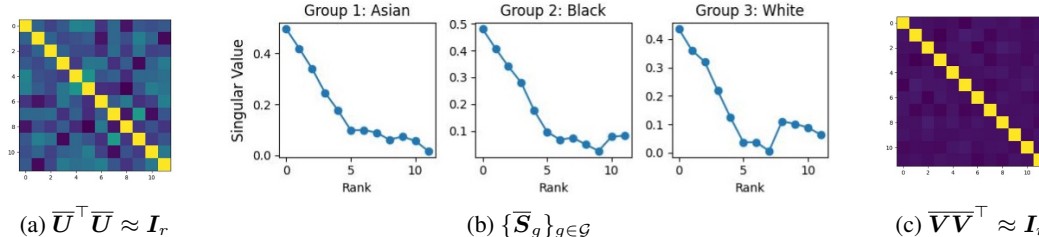

(a) $\overline{\boldsymbol{U}}^\top \overline{\boldsymbol{U}} \approx \boldsymbol{I}_r$        (b) $\{\overline{\boldsymbol{S}}_g\}_{g \in \mathcal{G}}$        (c) $\overline{\boldsymbol{V}}\overline{\boldsymbol{V}}^\top \approx \boldsymbol{I}_r$

Figure 4: Matrix visualization trained on **Race** attribute with ViT-B backbone (rank $r = 12$).

left and right singular vector matrices, and $\overline{\boldsymbol{S}}_g = \sum_{k=1}^{K} \alpha_{k,g} \boldsymbol{S}_{k,g}$ is the averaged fairness-aware singular value matrix. Here, $\alpha_{k,g} = n_{k,g}/n_g$ represents the weight for group $g$ at the client $k$, where $n_{k,g}$ is the number of data samples in group $g$ at client $k$, and $n_g = \sum_{g \in \mathbf{G}} n_{k,g}$ is the total number of samples in the group $g$ across all clients. The complete algorithm is detailed in Algorithm 1.

In FairLoRA, customized matrices $\boldsymbol{S}_{k,g}^t$ are specific to each demographic group, allowing the model to preserve unique characteristics within those groups. Moreover, the global shared parameters $\overline{\boldsymbol{S}}_g^t$ are calculated by averaging the client-specific matrices, weighted by the proportion $\alpha_{k,g}$ of data each client contributes to the global model for each demographic group. This ensures that the model is trained fairly, considering the distribution of data across different groups and clients.

**Initialization.** The initialization of the low-rank adaptation matrices in FairLoRA follows a structured approach to align with the SVD framework. The left singular matrix $\overline{\boldsymbol{U}}^0$ is initialized to zeros to provide a neutral starting point for local updates. The right singular matrix $\overline{\boldsymbol{V}}^0$ is initialized using a normal distribution, allowing for a diverse range of values to support efficient convergence during training. In addition, the singular value matrices $\{\overline{\boldsymbol{S}}_g^0\}_{g \in \mathcal{G}}$ is initialized using a linear space of values ranging from 0.5 to 0.1. For all groups, the first half of the ranks is initialized uniformly with this same linear space, ensuring that different demographic groups share the same principal singular vectors. For the remaining half of the ranks, a cyclic pattern is used to initialize the singular values, ensuring that each demographic group has its strongest response at distinct ranks. This design preserves group-specific diversity while enhancing the fairness across different groups. Fig. 4 illustrates the visualization of the matrices in FairLoRA. We observe that $\overline{\boldsymbol{U}}^\top \overline{\boldsymbol{U}} \approx \overline{\boldsymbol{V}}\overline{\boldsymbol{V}}^\top \approx \boldsymbol{I}_r$, indicating that both $\overline{\boldsymbol{U}}$ and $\overline{\boldsymbol{V}}$ form two sets of orthonormal bases. The singular value diagonal matrix $\overline{\boldsymbol{S}}_g$ has similar scales for the main singular values to capture common knowledge across demographic features, while exhibiting different response patterns for the smaller singular values to characterize the unique characteristics within the group.

---

**Algorithm 1** FairLoRA: Fairness-Aware Low-Rank Adaptation for Federated Learning

---

1: **Input:**
2:     $K$: number of clients,    $\{D_k\}_{k=1}^K$: local datasets for each client
3:     $\boldsymbol{W}_0$: pre-trained model weights of CLIP,    $\mathcal{G}$: set of demographic groups
4:     $\{\alpha_k\}_{k=1}^K, \{\alpha_{k,g}\}_{k,g=1,1}^{K,|\mathcal{G}|}$: weights for each client/group based on data contribution
5: **Initialization:**
6:     Initialize global parameters: $\overline{\boldsymbol{U}}^0 = 0, \overline{\boldsymbol{V}}^0 \sim \mathbf{N}(0,1), \{\overline{\boldsymbol{S}}_g^0\}_{g \in \mathcal{G}}$
7: **for** each round $t = 1, 2, \ldots, T$ **do**
8:     **Local Training:**
9:     **for** probability select clients $k \in \{1, 2, \ldots, K\}$ **do**
10:       **for** each iter $i = 1, 2, \ldots, |D_k|$ **do**
11:          $\Delta \boldsymbol{W}_k^t = \Delta \overline{\boldsymbol{W}}^{t-1} - \eta \nabla \mathcal{L}_{D_k^i}(\Delta \overline{\boldsymbol{W}}^{t-1})$, where $\Delta \overline{\boldsymbol{W}}^{t-1} = \overline{\boldsymbol{U}}^{t-1} (\sum_{g \in \mathcal{G}} \pi_g \overline{\boldsymbol{S}}_g^{t-1}) \overline{\boldsymbol{V}}^{t-1\top}$
12:       **end for**
13:       Retrieval local parameters: $\boldsymbol{U}_k^t, \boldsymbol{S}_{k,g}^t, \boldsymbol{V}_k^t \Leftarrow \Delta \boldsymbol{W}_k^t$
14:     **end for**
15:     **Global Aggregation:**
16:       Update global parameters: $\overline{\boldsymbol{U}}^t = \sum_{k=1}^K \alpha_k \boldsymbol{U}_k^t, \quad \overline{\boldsymbol{V}}^t = \sum_{k=1}^K \alpha_k \boldsymbol{V}_k^t, \quad \overline{\boldsymbol{S}}_g^t = \sum_{k=1}^K \alpha_{k,g} \boldsymbol{S}_{k,g}^t$
17: **end for**
18: Output global parameters: $\overline{\boldsymbol{W}}^T = \boldsymbol{W}_0 + \Delta \overline{\boldsymbol{W}}^T = \boldsymbol{W}_0 + \overline{\boldsymbol{U}}^T (\sum_{g \in \mathcal{G}} \pi_g \overline{\boldsymbol{S}}_g^T) \overline{\boldsymbol{V}}^{T\top}$

---

## 5 EXPERIMENTS

### 5.1 EXPERIMENTAL SETUP

**Implementation Details.** The training process consists of 50 epochs with a batch size of 32. The optimization is performed using Stochastic Gradient Descent (SGD) with a learning rate of 0.001, which decays by a factor of 0.1 at the 40th epoch. In each round, two out of three sites are randomly selected to update their local model weights before aggregating to form the global model. FairLoRA is implemented using two representative backbones: ResNet50 (He et al., 2016) and ViT-B (Dosovitskiy et al., 2020). For the ViT-B backbone, the rank and alpha parameters are set to 12 and 2, respectively, while for the ResNet50 backbone, they are set to 32 and 8. Additionally, all batch normalization layers in the ResNet50 architecture are unfrozen. To ensure training stability, we employ the exponential moving average (EMA) strategy to update the global parameters.

**Comparison Methods and Evaluation Metrics.** We compare FairLoRA with traditional fully parameter-updated methods, including FedAvg (McMahan et al., 2017) and FedHEAL (Chen et al., 2024), as well as prompt learning-based models, such as PromptFL (Guo et al., 2023) and FedOTP (Li et al., 2024), both based on CLIP (Radford et al., 2021). Compared to prompt learning models, traditional fully parameter-updated methods typically involve a larger number of learnable parameters. To facilitate the assessment of model fairness, we report the overall Area Under the Curve (AUC), equality-scale AUC (ESAUC) (Luo et al., 2024), and group-wise AUCs across the attributes of race, language and ethnicity.

### 5.2 BENCHMARK: FAIRFEDMED ON 2D SLO FUNDUS IMAGES

The experimental results in Tables 1 and 2 demonstrate the performance and fairness of various FL models applied to 2D SLO Fundus images for glaucoma detection within our proposed **FairFedMed** benchmark. Traditional FL methods such as *FedAvg* and *FedHEAL*, although effective in achieving overall AUCs ranging from 73% to 76%, exhibit fluctuating ESAUC scores between 57% and 68%. Analysis of group-wise AUCs reveals that these methods fall short in ensuring fairness across different demographic group. Prompt learning models, such as *PromptFL* and *FedOTP*, exhibit relatively lower performance with overall AUCs ranging from 71% to 73%. This decline is primarily due to the frozen model weights of CLIP, pre-trained on natural images, which do not adapt well to medical imaging data. Furthermore, these models struggle to achieve equitable outcomes across different demographic groups. For instance, for the *language* attribute, which includes a more diverse range of demographic groups, these models achieve only 56% to 68% ESAUC.

In contrast, *FairLoRA* consistently outperforms other models, delivering higher overall AUC scores and significantly enhancing fairness across all groups. Specifically, for the *race* attribute, *FairLoRA* improves the overall AUC by 4.1% using ResNet50 and by 3.4% using ViT-B. It also shows significant gains in ESAUC, with increases of 7.8% for ResNet50 and 5.9% for ViT-B. For the *language* attribute, *FairLoRA* achieves a significant 11.1% improvement in average ESAUC using the ResNet50 and a 4.7% improvement with the ViT-B, indicating the model's effectiveness in enhancing algorithmic fairness. For the *ethnicity* attribute, our model also achieves the highest ESAUC scores with both backbones. It is noteworthy that our model not only significantly improves fairness but also enhances the overall classification AUC by a substantial margin, with improvements of 4.7% with ResNet50 and 2.7% with ViT-B.

Overall, FairLoRA achieves state-of-the-art performance across all metrics with both ResNet50 and ViT-B, except for the *Others* group in the language attribute when using ResNet50. This indicates that our model consistently attains superior performance in terms of both accuracy and fairness.

### 5.3 BENCHMARK: FAIRFEDMED ON 3D OCT B-SCAN IMAGES

Tables 3 and 4 present the performance and fairness of various FL models on 3D OCT B-Scan images for glaucoma detection within our **FairFedMed** benchmark. Traditional FL methods, such as *FedAvg* and *FedHEAL*, achieve overall AUCs of 74% to 78% but exhibit fluctuating lower ESAUC scores between 65% and 70%. This gap between the overall AUC and ESAUC indicates challenges in ensuring fairness across demographic groups, particularly among the *Asian* population.

| Attribute | | Race | | | | | Language | | | | | Ethnicity | | | |
|---|---|---|---|---|---|---|---|---|---|---|---|---|---|---|---|
| Model | Client ID | Overall AUC | ES AUC | Asian AUC | Black AUC | White AUC | Overall AUC | ES AUC | English AUC | Spanish AUC | Others AUC | Overall AUC | ES AUC | NonHisp. AUC | Hisp. AUC |
| *Fully parameter-updated FL models* | | | | | | | | | | | | | | | |
| FedAvg | C1 | 73.5 | 67.8 | 78.3 | 70.7 | 72.7 | 73.9 | 54.7 | 79.6 | 58.3 | 87.6 | 72.3 | 63.14 | 79.75 | 65.3 |
| | C2 | 74.9 | 66.3 | 79.2 | 69.2 | 71.4 | 74.3 | 62.4 | 72.5 | 68.9 | 62.3 | 72.6 | 68.7 | 75.3 | 69.3 |
| | C3 | 72.8 | 67.8 | 76.9 | 71.8 | 70.3 | 72.3 | 54.1 | 69.7 | 80.3 | 95.3 | 70.8 | 68.8 | 71.6 | 72.9 |
| | Avg. | 73.7 | 67.8 | 78.1 | 70.7 | 71.5 | 73.5 | 57.1 | 73.9 | 69.2 | 81.7 | 71.8 | 66.9 | 75.6 | 69.3 |
| FedHEAL | C1 | 74.4 | 66.2 | 79.4 | 69.3 | 71.9 | 73.5 | 57.3 | 81.6 | 65.5 | 85.7 | 73.6 | 66.8 | 77.8 | 67.4 |
| | C2 | 74.2 | 64.4 | 82.3 | 71.8 | 69.5 | 74.9 | 64.4 | 74.2 | 65.3 | 68.9 | 71.8 | 58.8 | 78.6 | 56.3 |
| | C3 | 73.4 | 68.2 | 75.9 | 70.2 | 71.3 | 75.0 | 58.3 | 72.8 | 81.6 | 94.2 | 72.8 | 70.1 | 73.4 | 69.6 |
| | Avg. | 74.1 | 66.3 | 79.2 | 70.5 | 70.9 | 74.5 | 60.0 | 76.2 | 70.8 | **82.9** | 72.7 | 65.3 | 76.5 | 64.5 |
| *Prompt learning-based FL models* | | | | | | | | | | | | | | | |
| PromptFL | C1 | 72.9 | 69.8 | 76.2 | 71.8 | 73.0 | 72.6 | 57.2 | 71.8 | 87.5 | 83.7 | 71.9 | 67.3 | 71.7 | 78.6 |
| | C2 | 71.4 | 66.0 | 78.0 | 70.4 | 70.8 | 71.8 | 62.8 | 72.3 | 67.1 | 62.7 | 72.2 | 64.3 | 73.0 | 60.5 |
| | C3 | 72.3 | 65.8 | 70.7 | 64.8 | 73.1 | 70.4 | 55.0 | 69.8 | 81.3 | 87.0 | 69.5 | 60.7 | 69.8 | 55.2 |
| | Avg. | 72.2 | 67.2 | 75.0 | 69.0 | 72.3 | 71.6 | 58.3 | 71.3 | 78.6 | 77.8 | 71.2 | 64.1 | 71.5 | 64.8 |
| FedOTP | C1 | 72.1 | 66.8 | 75.7 | 67.9 | 72.4 | 72.2 | 54.2 | 71.6 | 93.8 | 83.1 | 71.9 | 64.7 | 78.2 | 71.5 |
| | C2 | 71.0 | 64.8 | 77.9 | 68.8 | 70.5 | 71.6 | 61.3 | 72.5 | 66.3 | 61.0 | 72.2 | 66.5 | 76.3 | 72.2 |
| | C3 | 70.8 | 64.1 | 73.6 | 63.8 | 71.4 | 70.6 | 52.8 | 70.0 | 78.1 | 96.3 | 69.1 | 59.5 | 66.1 | 69.5 |
| | Avg. | 71.3 | 65.2 | 75.7 | 66.8 | 71.4 | 71.5 | 56.1 | 71.4 | 79.4 | 80.1 | 71.1 | 63.6 | 73.5 | 71.1 |
| FairLoRA | C1 | 78.1 | 76.7 | 77.1 | 78.7 | 77.8 | 77.3 | 68.1 | 77.6 | 72.7 | 68.7 | 78.0 | 73.7 | 78.1 | 72.3 |
| | C2 | 78.4 | 74.0 | 79.4 | 73.6 | 78.7 | 77.8 | 70.6 | 77.9 | 86.2 | 79.4 | 76.6 | 71.1 | 77.1 | 69.3 |
| | C3 | 78.0 | 76.0 | 78.1 | 75.6 | 77.9 | 78.6 | 74.5 | 78.6 | 83.3 | 79.5 | 77.6 | 77.2 | 77.7 | 77.1 |
| | Avg. | **78.2** | **75.6** | **78.2** | **76.0** | **78.1** | **77.9** | **71.1** | **78.0** | **80.7** | 75.9 | **77.4** | **74.0** | **77.6** | **72.9** |
| | Δ Avg. | +4.1 | +7.8 | +0.1 | +5.3 | +5.8 | +3.4 | +11.1 | +1.8 | +1.3 | -7.0 | +4.7 | +7.1 | +1.1 | +1.8 |

Table 1: Quantitative comparison and fairness analysis conducted on **2D SLO Fundus** images, trained with the **ResNet50** backbone. 'Hisp.' is an abbreviation for 'Hispanic'.

| Attribute | | Race | | | | | Language | | | | | Ethnicity | | | |
|---|---|---|---|---|---|---|---|---|---|---|---|---|---|---|---|
| Model | Client ID | Overall AUC | ES AUC | Asian AUC | Black AUC | White AUC | Overall AUC | ES AUC | English AUC | Spanish AUC | Others AUC | Overall AUC | ES AUC | NonHisp. AUC | Hisp. AUC |
| *Fully parameter-updated FL models* | | | | | | | | | | | | | | | |
| FedAvg | C1 | 75.7 | 69.1 | 79.3 | 69.9 | 75.5 | 76.2 | 56.6 | 75.2 | 52.8 | 86.5 | 77.1 | 72.3 | 79.2 | 72.6 |
| | C2 | 74.6 | 65.7 | 81.5 | 78.6 | 71.9 | 75.8 | 67.6 | 76.2 | 75.7 | 64.3 | 78.9 | 67.6 | 78.2 | 62.7 |
| | C3 | 74.2 | 68.5 | 78.7 | 73.3 | 71.2 | 73.2 | 62.9 | 74.3 | 70.9 | 60.3 | 71.7 | 64.5 | 73.1 | 62.1 |
| | Avg. | 74.8 | 67.7 | 79.8 | 73.0 | 72.9 | 75.1 | 62.4 | 75.3 | 66.5 | 70.4 | 75.9 | 68.1 | 76.9 | 65.8 |
| FedHEAL | C1 | 76.7 | 69.2 | 80.2 | 70.3 | 77.6 | 75.3 | 55.8 | 76.3 | 53.5 | 87.6 | 76.9 | 72.5 | 78.2 | 72.2 |
| | C2 | 75.1 | 66.3 | 80.3 | 79.6 | 71.5 | 76.5 | 67.3 | 75.5 | 74.9 | 65.3 | 77.5 | 67.5 | 79.7 | 64.9 |
| | C3 | 75.9 | 67.8 | 79.8 | 76.3 | 68.2 | 73.9 | 64.1 | 74.3 | 71.5 | 61.4 | 73.1 | 64.9 | 75.2 | 62.5 |
| | Avg. | 75.9 | 67.8 | 80.1 | **75.4** | 72.4 | 75.2 | 62.4 | 75.4 | 66.6 | 71.4 | 75.8 | 68.3 | 77.7 | 66.6 |
| *Prompt learning-based FL models* | | | | | | | | | | | | | | | |
| PromptFL | C1 | 73.4 | 67.4 | 77.8 | 69.2 | 73.5 | 74.1 | 54.4 | 73.7 | 50.0 | 85.8 | 71.5 | 71.4 | 71.5 | 71.4 |
| | C2 | 73.6 | 66.1 | 80.3 | 76.1 | 71.6 | 74.2 | 63.2 | 74.9 | 69.0 | 62.6 | 73.6 | 60.6 | 75.0 | 53.4 |
| | C3 | 73.6 | 68.5 | 77.0 | 69.6 | 73.5 | 72.0 | 56.9 | 72.5 | 93.8 | 51.9 | 69.1 | 63.1 | 69.5 | 59.9 |
| | Avg. | 73.5 | 67.3 | 78.4 | 71.6 | 72.9 | 73.4 | 56.1 | 73.4 | 70.9 | 66.8 | 71.4 | 65.0 | 72.0 | 61.6 |
| FedOTP | C1 | 71.9 | 66.3 | 77.7 | 69.3 | 71.9 | 73.4 | 54.6 | 72.8 | 53.1 | 86.9 | 72.3 | 71.3 | 72.3 | 73.7 |
| | C2 | 72.4 | 66.2 | 78.5 | 74.1 | 70.7 | 72.1 | 67.2 | 72.3 | 72.0 | 65.1 | 74.0 | 60.5 | 75.4 | 53.2 |
| | C3 | 72.5 | 65.9 | 75.4 | 65.6 | 72.7 | 69.8 | 46.0 | 69.4 | 93.8 | 42.6 | 68.4 | 61.1 | 68.7 | 56.8 |
| | Avg. | 72.3 | 66.1 | 77.2 | 69.7 | 71.8 | 71.8 | 56.0 | 71.5 | 73.0 | 64.9 | 71.6 | 64.3 | 72.1 | 61.2 |
| FairLoRA | C1 | 78.3 | 73.5 | 79.1 | 73.5 | 77.4 | 77.3 | 65.4 | 77.5 | 63.6 | 73.0 | 78.6 | 75.9 | 78.7 | 82.1 |
| | C2 | 79.9 | 73.7 | 80.6 | 72.9 | 80.5 | 78.2 | 70.2 | 78.1 | 70.4 | 81.6 | 77.9 | 71.8 | 78.5 | 70.0 |
| | C3 | 79.7 | 73.8 | 83.5 | 75.8 | 79.4 | 79.4 | 65.7 | 79.1 | 95.8 | 83.3 | 79.3 | 70.7 | 79.3 | 91.4 |
| | Avg. | **79.3** | **73.7** | **81.1** | 74.1 | **79.1** | **78.3** | **67.1** | **78.2** | **76.6** | **79.3** | **78.6** | **72.8** | **78.8** | **81.2** |
| | Δ Avg. | +3.4 | +5.9 | +1.0 | -1.3 | +6.2 | +3.1 | +4.7 | +2.8 | +3.6 | +7.9 | +2.7 | +4.5 | +1.1 | +14.6 |

Table 2: Quantitative comparison and fairness analysis conducted on **2D SLO Fundus** images, trained with the **ViT-B/16** backbone. 'Hisp.' is an abbreviation for 'Hispanic'.

Prompt-based models like *PromptFL* and *FedOTP* display overall AUCs ranging from 73% to 78%. Although they occasionally achieve higher group-specific AUCs – for example, *FedOTP* reaches 79.6% for the *Asian* group using ResNet50, they generally fail to maintain consistent fairness across all demographics. These prompt-based models attain ESAUC scores between 65% and 71%, reflecting significant gaps relative to overall AUC and indicating inequitable outcomes across groups.

In contrast, *FairLoRA* consistently outperforms other models, delivering higher overall AUCs and significantly enhancing fairness across all groups. For the *race* attribute, it achieves overall AUC

| Attribute | | Race | | | | | Language | | | | | Ethnicity | | | |
|---|---|---|---|---|---|---|---|---|---|---|---|---|---|---|---|
| Model | Client ID | Overall AUC | ES AUC | Asian AUC | Black AUC | White AUC | Overall AUC | ES AUC | English AUC | Spanish AUC | Others AUC | Overall AUC | ES AUC | NonHisp. AUC | Hisp. AUC |
| *Fully parameter-updated FL models* | | | | | | | | | | | | | | | |
| FedAvg | C1 | 75.2 | 68.6 | 71.9 | 77.4 | 79.3 | 76.9 | 71.3 | 73.2 | 78.5 | 79.3 | 75.8 | 69.2 | 73.6 | 83.3 |
| | C2 | 72.6 | 69.3 | 75.3 | 71.3 | 73.5 | 74.7 | 71.8 | 74.3 | 77.8 | 75.3 | 76.2 | 74.6 | 75.2 | 74.9 |
| | C3 | 74.6 | 63.2 | 65.4 | 79.6 | 78.3 | 74.0 | 62.7 | 69.3 | 77.5 | 83.7 | 74.9 | 67.3 | 75.2 | 83.8 |
| | Avg. | 74.1 | 67.0 | 70.9 | 76.1 | 77.0 | 75.2 | 68.6 | 72.3 | 78.0 | 79.4 | 75.7 | 70.4 | 75.5 | 80.7 |
| FedHEAL | C1 | 76.5 | 70.1 | 72.8 | 78.1 | 80.2 | 77.4 | 71.6 | 75.3 | 80.6 | 80.2 | 74.2 | 67.1 | 73.6 | 84.1 |
| | C2 | 75.4 | 71.8 | 73.6 | 72.5 | 75.6 | 74.9 | 70.4 | 72.8 | 79.0 | 74.7 | 75.9 | 72.0 | 77.5 | 77.2 |
| | C3 | 74.2 | 61.4 | 64.4 | 79.7 | 79.7 | 74.2 | 65.4 | 73.8 | 65.3 | 78.3 | 77.1 | 67.4 | 73.4 | 87.9 |
| | Avg. | 75.3 | 67.8 | 70.3 | 76.8 | 78.5 | 75.5 | 69.1 | 73.9 | 75.0 | 77.7 | 75.7 | 68.8 | 74.9 | 81.4 |
| *Prompt learning-based FL models* | | | | | | | | | | | | | | | |
| PromptFL | C1 | 74.3 | 72.3 | 71.8 | 74.5 | 74.4 | 75.9 | 68.6 | 75.8 | 81.8 | 80.6 | 73.1 | 65.5 | 72.8 | 84.4 |
| | C2 | 71.9 | 67.7 | 76.7 | 70.7 | 72.1 | 71.7 | 68.3 | 74.9 | 72.9 | 71.1 | 72.1 | 71.7 | 72.0 | 72.7 |
| | C3 | 73.1 | 66.6 | 82.6 | 73.0 | 73.1 | 71.3 | 67.0 | 71.2 | 66.7 | 73.1 | 74.7 | 68.3 | 74.5 | 83.8 |
| | Avg. | 73.1 | 68.9 | 77.0 | 72.7 | 73.2 | 73.0 | 68.0 | 74.0 | 73.8 | 74.9 | 73.3 | 68.5 | 73.1 | 80.3 |
| FedOTP | C1 | 74.5 | 72.2 | 72.6 | 75.8 | 74.4 | 74.7 | 72.8 | 74.7 | 75.0 | 77.0 | 72.6 | 63.2 | 72.2 | 87.1 |
| | C2 | 72.3 | 65.7 | 79.1 | 69.4 | 72.6 | 72.7 | 68.7 | 72.7 | 78.5 | 72.8 | 74.0 | 72.7 | 74.1 | 72.3 |
| | C3 | 73.5 | 64.5 | 87.1 | 73.1 | 73.5 | 72.0 | 64.4 | 72.1 | 62.5 | 74.4 | 75.9 | 66.6 | 75.5 | 89.5 |
| | Avg. | 73.4 | 67.5 | 79.6 | 72.8 | 73.5 | 73.1 | 68.6 | 73.2 | 72.0 | 74.7 | 74.2 | 67.5 | 73.9 | 83.0 |
| FairLoRA | C1 | 78.1 | 76.7 | 77.1 | 78.7 | 77.8 | 77.3 | 66.8 | 77.7 | 77.3 | 62.1 | 78.3 | 72.0 | 78.4 | 69.6 |
| | C2 | 78.4 | 74.0 | 79.4 | 73.6 | 78.7 | 77.4 | 72.6 | 77.5 | 72.3 | 76.1 | 76.4 | 66.1 | 77.1 | 61.6 |
| | C3 | 78.0 | 76.0 | 78.1 | 75.6 | 77.9 | 80.7 | 67.6 | 80.5 | 95.8 | 84.6 | 77.4 | 74.0 | 77.3 | 81.9 |
| | Avg. | 78.2 | 75.6 | 78.2 | 76.0 | 78.1 | 78.5 | 69.0 | 78.6 | 81.8 | 74.3 | 77.4 | 70.7 | 77.6 | 71.0 |
| | Δ Avg. | +2.9 | +6.7 | -1.4 | -0.8 | -0.4 | +3.0 | -0.1 | +4.6 | +3.8 | -5.1 | +1.7 | +0.3 | +2.1 | -12.0 |

Table 3: Quantitative comparison and fairness analysis conducted on **3D OCT B-Scan** images trained with the **ResNet50** backbone. 'Hisp.' is an abbreviation for 'Hispanic'.

| Attribute | | Race | | | | | Language | | | | | Ethnicity | | | |
|---|---|---|---|---|---|---|---|---|---|---|---|---|---|---|---|
| Model | Client ID | Overall AUC | ES AUC | Asian AUC | Black AUC | White AUC | Overall AUC | ES AUC | English AUC | Spanish AUC | Others AUC | Overall AUC | ES AUC | NonHisp. AUC | Hisp. AUC |
| *Fully parameter-updated FL models* | | | | | | | | | | | | | | | |
| FedAvg | C1 | 76.4 | 69.5 | 71.6 | 74.2 | 79.3 | 76.7 | 60.7 | 78.2 | 89.7 | 88.5 | 77.2 | 75.3 | 77.2 | 74.7 |
| | C2 | 75.3 | 66.6 | 83.0 | 71.5 | 73.8 | 76.9 | 68.1 | 80.3 | 74.9 | 69.3 | 76.7 | 67.1 | 78.2 | 63.8 |
| | C3 | 77.6 | 69.4 | 81.2 | 72.6 | 74.4 | 75.5 | 65.7 | 75.3 | 85.3 | 80.3 | 78.3 | 65.0 | 74.8 | 95.3 |
| | Avg. | 76.4 | 68.5 | 78.6 | 72.8 | 75.8 | 76.4 | 64.8 | 77.9 | 83.3 | 79.4 | 77.4 | 69.1 | 76.7 | 77.9 |
| FedHEAL | C1 | 77.8 | 70.9 | 72.6 | 73.7 | 78.2 | 79.2 | 64.8 | 76.9 | 89.6 | 88.7 | 78.3 | 73.8 | 78.6 | 72.5 |
| | C2 | 78.3 | 66.5 | 85.9 | 71.8 | 74.6 | 78.8 | 71.9 | 80.3 | 79.7 | 71.5 | 77.3 | 66.7 | 80.2 | 64.5 |
| | C3 | 78.7 | 66.9 | 83.9 | 69.2 | 75.7 | 77.9 | 67.9 | 76.3 | 88.3 | 75.2 | 79.3 | 70.2 | 76.7 | 89.8 |
| | Avg. | 78.3 | 68.1 | 80.8 | 71.6 | 76.2 | 78.6 | 68.2 | 77.8 | 85.9 | 78.5 | 78.3 | 70.3 | 78.5 | 75.6 |
| *Prompt learning-based FL models* | | | | | | | | | | | | | | | |
| PromptFL | C1 | 77.8 | 62.6 | 61.7 | 71.2 | 79.4 | 78.3 | 67.3 | 78.1 | 90.9 | 81.9 | 77.2 | 77.2 | 77.2 | 77.2 |
| | C2 | 78.4 | 68.3 | 86.0 | 71.7 | 78.9 | 77.7 | 69.4 | 78.3 | 73.1 | 70.9 | 75.7 | 68.9 | 76.1 | 66.3 |
| | C3 | 77.9 | 68.1 | 84.4 | 70.4 | 78.9 | 76.9 | 64.2 | 76.5 | 87.5 | 85.9 | 79.7 | 66.5 | 79.1 | 99.1 |
| | Avg. | 78.0 | 66.3 | 77.4 | 71.1 | 78.9 | 77.6 | 67.0 | 77.7 | 83.8 | 79.6 | 77.5 | 70.9 | 77.5 | 80.9 |
| FedOTP | C1 | 77.6 | 63.3 | 61.5 | 72.5 | 79.0 | 76.5 | 68.8 | 76.3 | 81.8 | 82.1 | 76.1 | 73.4 | 76.2 | 72.5 |
| | C2 | 76.7 | 67.1 | 83.6 | 69.7 | 77.0 | 76.2 | 66.0 | 77.1 | 71.0 | 66.8 | 73.8 | 68.1 | 74.2 | 65.8 |
| | C3 | 76.7 | 67.5 | 84.3 | 70.9 | 76.9 | 74.7 | 60.7 | 74.4 | 87.5 | 84.6 | 78.3 | 65.1 | 77.8 | 98.1 |
| | Avg. | 77.0 | 66.0 | 76.5 | 71.0 | 77.6 | 75.8 | 65.2 | 75.9 | 80.1 | 77.8 | 76.1 | 68.9 | 76.1 | 78.8 |
| FairLoRA | C1 | 84.5 | 72.0 | 73.4 | 78.9 | 85.1 | 82.1 | 73.9 | 81.7 | 75.0 | 85.6 | 82.3 | 79.7 | 82.6 | 79.2 |
| | C2 | 82.4 | 71.6 | 90.5 | 75.5 | 82.2 | 81.5 | 76.8 | 81.8 | 79.2 | 77.9 | 80.7 | 71.8 | 81.5 | 69.2 |
| | C3 | 84.0 | 74.3 | 89.6 | 76.5 | 84.1 | 82.5 | 73.7 | 82.5 | 87.5 | 75.6 | 82.7 | 73.8 | 82.3 | 94.3 |
| | Avg. | 83.6 | 72.6 | 84.5 | 77.0 | 83.8 | 82.0 | 74.8 | 82.0 | 80.6 | 79.7 | 81.9 | 75.1 | 82.1 | 80.9 |
| | Δ Avg. | +5.3 | +4.1 | +3.7 | +4.2 | +4.9 | +3.4 | +6.6 | +4.1 | -5.3 | +0.1 | +3.6 | +4.2 | +3.6 | +0.0 |

Table 4: Quantitative comparison and fairness analysis conducted on **3D OCT B-Scan** images, trained with the **ViT-B/16** backbone. 'Hisp.' is an abbreviation for 'Hispanic'.

increases of 2.9% (ResNet50) and 5.3% (ViT-B), with ESAUC gains of 6.7% and 4.1%, respectively. Regarding the *language* attribute, *FairLoRA* improves average ESAUC by 6.6% (ViT-B), highlighting its effectiveness in enhancing algorithmic fairness. While slight decreases occur for certain groups, such as a 1.4% drop for the *Asian* group using ResNet50, this is primarily due to LoRA's limitations when applied to convolutional architectures.

Notably, *FairLoRA* enhances not only fairness but also overall classification AUC by significant margins, 3.4% to 5.3% with ViT-B and 1.7% to 3.0% with ResNet50, demonstrating SOTA perfor-

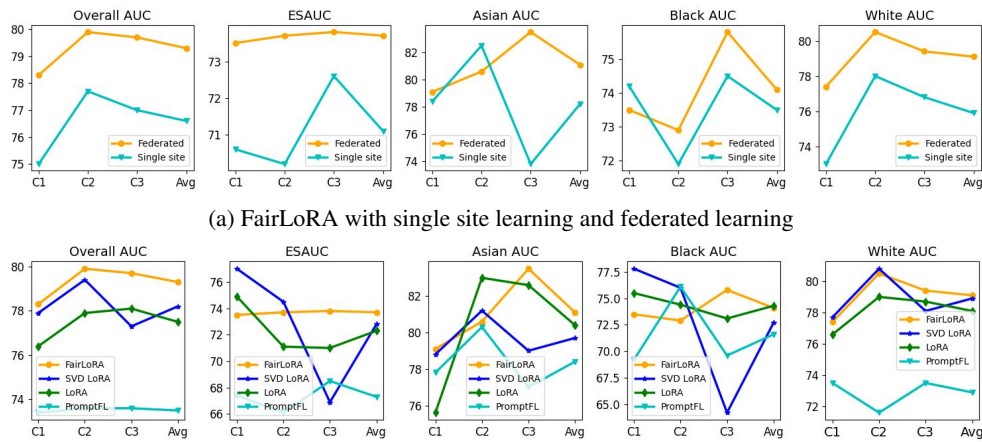

(a) FairLoRA with single site learning and federated learning

(b) Baseline model and three LoRA variants: LoRA, SVD-based LoRA and FairLoRA

Figure 5: Ablation study conducted on the **Race** attribute of **2D SLO Fundus** images.

mance across all metrics. This consistent improvement in both accuracy and fairness for 3D imaging demonstrates *FairLoRA*'s significant effectiveness in federated learning for medical imaging, particularly in handling diverse demographic groups across both 2D and 3D imaging modalities.

### 5.4 ABLATION STUDY

Ablation studies are conducted on the **Race** attribute of **2D SLO Fundus** images, using the ViT-B backbone. Please refer to the **Appendix B** for more ablation studies.

**Single Site Learning VS. Federated Learning.** The ablation study shown in Fig. 5a compares single-client learning and federated learning using the FairLoRA model. Federated learning consistently outperforms single site learning in terms of overall AUC and ESAUC across all clients, with improvements of 1%-3%. Federated learning achieves better performance on most group-wise AUCs, demonstrating its ability to enhance fairness and overall model accuracy across different demographic groups compared to single site learning.

**Three LoRA variants.** Fig. 5b provides a comparative analysis of the baseline model, PromptFL, alongside three LoRA variants: *LoRA*, *SVD-based LoRA*, and *FairLoRA*. PromptFL demonstrates the lowest overall performance, with an average overall AUC and fairness metric ESAUC of only around 73% and 67%, respectively. Introducing *LoRA* for model fine-tuning shows significant improvements across most metrics, achieving the overall AUC of 77.5% and the ESAUC of 72%. The *SVD-based LoRA* model maintains robust overall performance. However, its fairness metrics ESAUC and group-wise AUCs show significant fluctuations across clients, indicating an inability to effectively handle distribution differences between clients. In contrast, our proposed *FairLoRA* outperforms other models in overall AUC, achieving the highest score of 79.3%. Additionally, it surpasses all models in both ESAUC and group-wise AUCs across the average metrics of the three clients, demonstrating the best fairness performance. Overall, *FairLoRA* shows exceptional capability in medical image classification and effectively balances fairness across different demographics.

## 6 CONCLUSION

This paper addresses the lack of fairness-aware medical datasets in FL by introducing the FairFedMed dataset, which is the first to include both 2D and 3D real-world medical images alongside key demographic attributes. We also propose FairLoRA, a novel framework designed to enhance fairness through a low-rank approximation method tailored for demographic groups. FairLoRA ensures that both intra-group and inter-group characteristics are preserved, and global knowledge is shared across clients. Our experimental results confirm that FairLoRA not only achieves superior performance in medical image classification but also promotes fairness across different demographic groups, making a significant contribution to fairness-aware FL in healthcare.

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
