# Appendix to "FairFedMed: Achieving Equity in Medical Federated Learning via FairLoRA"

In this appendix, we provide the following materials:

A  Visualization of 2D SLO Fundus and 3D OCT B-Scan images for glaucoma classification (referring to Sec. 3 in the main paper);

B  More ablation study for FairLoRA (referring to Sec. 5.4 in the main paper);

## A  FEDFAIRMED DATASET

**2D SLO Fundus images.** 2D SLO Fundus images provide detailed visualizations of the retina, allowing for the assessment of structural changes associated with glaucoma. As shown in Fig. 1, these images capture critical features, such as the optic nerve head and retinal nerve fiber layer, which are essential for evaluating glaucomatous damage.

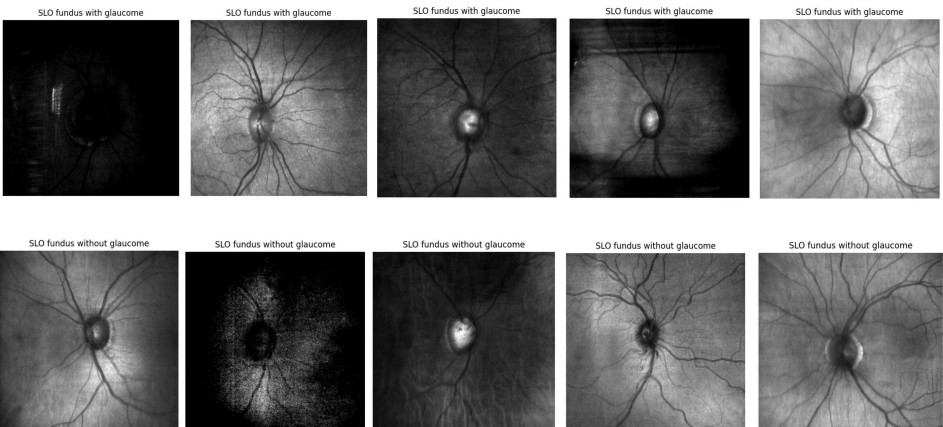

Figure 1: 2D SLO Fundus images with or without glaucoma disease

**3D OCT B-scan images.** By comprehensively assessing key factors such as retinal nerve fiber layer (RNFL) thickness, cup-to-disc ratio (C/D ratio), and macular structure, physicians can obtain clear diagnostic criteria for glaucoma from OCT B-scans. This high-resolution imaging method accurately captures subtle changes in ocular structures, aiding in the early identification of glaucoma and monitoring its progression. Due to the non-invasive and high-precision nature of OCT B-scans, they provide a reliable tool for the diagnosis and management of glaucoma, significantly enhancing the effectiveness of clinical interventions. Additionally, OCT B-scans generate 128 images, from which we select one image every four for visualization, as shown in Fig. 2.

## B  ABLATION STUDY FOR FAIRLORA

**Training stability.** Fig. 3a illustrates the overall AUC convergence across different models—FairLoRA, SVD LoRA, and LoRA—during training, with all model weights updated using exponential moving average (EMA). FairLoRA exhibits the most stable and superior performance, maintaining overall AUC around 79-80% after early convergence, with minimal fluctuations, demonstrating its robustness. In contrast, SVD LoRA achieves a comparable AUC performance but shows minor instability as its AUC slightly declines after reaching its peak, stabilizing between 77-78%. The LoRA model shows the most volatile behavior, with significant fluctuations throughout training and a failure to maintain consistent AUC performance, fluctuating between 72-76%. Overall, FairLoRA consistently outperforms the others, offering a clear advantage in both stability and accuracy during the training process for federated learning.

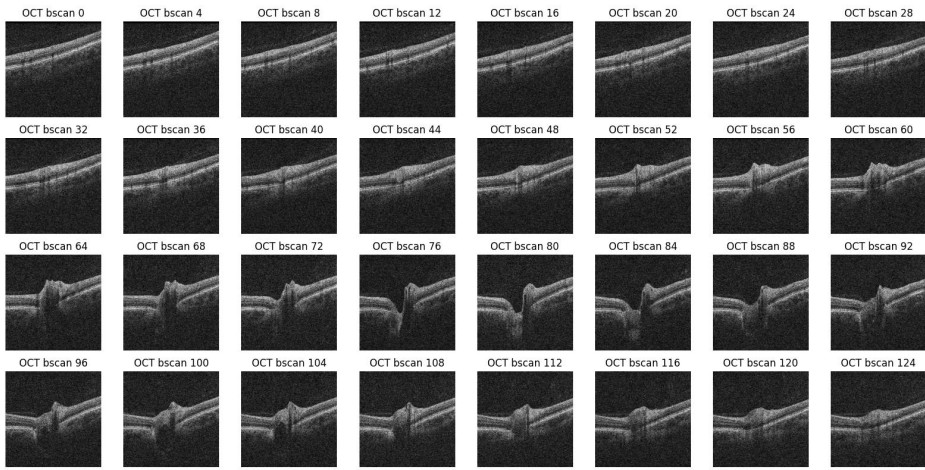

Figure 2: A visualization example of 3D OCT B-scan images depicting glaucoma.

**Initialization of group-wise singular values $\overline{S}_g^0$.** The impact of singular value initialization on overall AUC is shown in Fig. 3b. We can observe that the initialization with the same linear values across all demographic groups achieves the highest performance in the early epochs, reaching an AUC of approximately 79.5%, before subsequently declining to 77.5%. In contrast, the cyclic shift initialization method yields a slightly lower overall AUC, indicating potential inefficiencies in convergence. The half-half initialization exhibits the most stable performance, maintaining an AUC consistently around 79%, which suggests that it effectively balances the influence of different demographic groups during training.

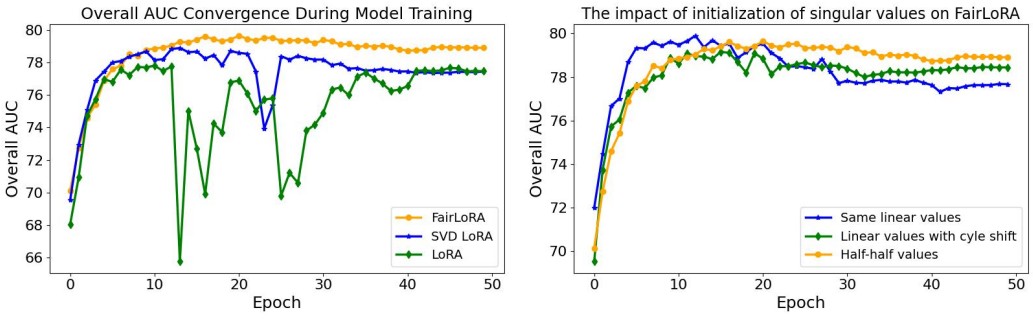

(a) Overall AUC convergence during model training     (b) Three initialization for singular vales

Figure 3: Ablation study on training convergence of three LoRA variants and the impact on singular values initialization . Here, 'Half-half values' refers to the initialization in Sec. 4.3 of the main paper.