# OpenReview forum: "FairFedMed: Achieving Equity in Medical Federated Learning via FairLoRA"
_ICLR.cc/2025/Conference — ICLR 2025 Conference Withdrawn Submission_

### Official Review · Reviewer_UU5X · 2024-10-29

**Soundness:** 2
**Presentation:** 3
**Contribution:** 3
**Rating:** 3
**Confidence:** 4

**Summary:**

The authors propose FairLoRA, a LoRA-like method for fair federated finetuning. The method works by performing LoRA-like fine-tuning updates with parts of the updates specific to different (pre-specified and labeled) demographic groups, and then aggregating those local updates globally. The result is a fine-tuned and group-customized global model trained on private data. In addition, a new Federated Learning (FL) benchmark dataset is contributed, which consists of 2D and 3D medical images. The proposed FairLoRA method is compared (using two CLIP base models) against several baselines on this dataset, performing very favorably both in terms of overall performance as well as group fairness metrics.

**Strengths:**

- The paper is well-written overall, easy to follow, and generally nicely presented.
- Fair FL approaches in the medical imaging domain are indeed under-appreciated, under-researched, and important.
- The proposed scheme is lightweight / efficient, relatively simple to implement, and performs very favorably on the newly contributed real-world medical image dataset(s).
- A new real-world medical image dataset is contributed - including multi-attribute demographic patient information, which is (sadly) rare and much appreciated.

**Weaknesses:**

- I don't (yet?) see how / why the main algorithm presented here should be a correct federated implementation of truncated SVD, as is claimed. This concerns in particular the global aggregation part, i.e. eq. (3): If $A \approx U_A S_A V_A^T$ and $B \approx U_B S_B V_B^T$ then it does not generally follow that $A + B \approx (U_A + U_B) (S_A + S_B) (V_A + V_B)^T$? Or am I missing something crucial here? (Intuitively, even if "similar" singular components are identified at the different sites - unclear why that would necessarily be the case - why should these also be in the same order?)
  - Also, the authors essentially propose a new method for performing federated SVD here, on which there is already research which is neither acknowledged nor discussed [1-3].
  - Finally, the whole premise of FL is that data remain private per site. This is not necessarily the case if full SVD updates are shared, hence the various proposals for privacy-preserving federated SVD. While I would imagine that this isn't a realistic concern here due to the low truncation order, an argument to this effect should at least be provided.
- No details beyond the pure data content are available about the new FairFedMed dataset. What is the source of the data? How were these recordings obtained? Do the authors have the rights to publish this dataset? Was consent obtained from the human subjects and / or the rights holders? Under which license is the dataset provided?
  - Also, the authors simply write that they "divide all subjects into three separate sites" - does that mean that these are not real sites? If not, what makes these 'synthetic' sites in any way similar to real sites, i.e., exhibiting realistic data shifts etc.? In other words, how is this different from an arbitrary medical imaging dataset that is randomly split into three parts?
- It remained unclear to me which model parts exactly the different compared methods are tuning, partially because the whole modeling setup remained unclear to me - it is simply not described at all. I can try to deduce some things from Fig. 3 but it's really not very clear. Which piece of information is fed into which model component in order to pass from the input image to the output prediction? What even *is* the output of the pipeline? And, again, which parts of this pipeline are tuned by the different methods? I only gathered from the tables that apparently FairLoRA only tunes a (soft) prompt? This is not at all apparent from the remainder of the text.
- Given that the whole premise of the paper and approach is that different models should be learned for different demographic groups, I would strongly suggest adding at least some discussion of situations in which this might be desirable or not. Cf. the extensive recent debates on race adjustments in clinical risk models [4-5]. Also see [6] on the risks of modeling based on coarse demographic data. This also raises questions of causality: is demographic group membership really a cause of the prediction target, or is it just a correlate of an underlying cause that is (as of yet) unmeasured (but maybe should be) [7]? This will have implications for generalizability across domains. Learning group-specific models also hugely increases the risk of learning demographic shortcuts and thereby reinforcing existing (e.g. underdiagnosis) biases [8]. Compare this to e.g. equal opportunity regularization, which can sometimes mitigate label biases [9].
- The authors claim that "Despite the urgent need for fairness in healthcare, research on group fairness in federated learning for medical applications is limited, revealing a significant gap in this field." However, there is actually quite a bit of research in this area that is not acknowledged, cf. e.g. the various refs in [10].

#### References
- [1] [Chai et al., Practical Lossless Federated Singular Vector Decomposition over Billion-Scale Data](https://dl.acm.org/doi/abs/10.1145/3534678.3539402)
- [2] [Hardtebrodt et al., Federated singular value decomposition for high-dimensional data](https://link.springer.com/article/10.1007/s10618-023-00983-z)
- [3] [Grammenos et al., Federated Principal Component Analysis](https://proceedings.neurips.cc/paper/2020/hash/47a658229eb2368a99f1d032c8848542-Abstract.html)
- [4] [Zink et al., Race adjustments in clinical algorithms can help correct for racial disparities in data quality](https://www.pnas.org/doi/full/10.1073/pnas.2402267121)
- [5] [Diao et al., Implications of Race Adjustment in Lung-Function Equations](https://pmc.ncbi.nlm.nih.gov/articles/PMC11305821/)
- [6] [Movva et al., Coarse race data conceals disparities in clinical risk score performance](https://proceedings.mlr.press/v219/movva23a.html)
- [7] [Petersen et al., The path toward equal performance in medical machine learning](https://www.cell.com/patterns/fulltext/S2666-3899(23)00145-9)
- [8] [Yang et al., The limits of fair medical imaging AI in real-world generalization](https://www.nature.com/articles/s41591-024-03113-4)
- [9] [Blum and Stangl, Recovering from Biased Data: Can Fairness Constraints Improve Accuracy?](https://arxiv.org/pdf/1912.01094)
- [10] [Chen et al., Algorithmic fairness in artificial intelligence for medicine and healthcare](https://www.nature.com/articles/s41551-023-01056-8)

**Questions:**

See above.

**Details Of Ethics Concerns:**

See above: concerning the newly provided dataset, no details are provided regarding its provenance, licensing / right to publish, or patient consent.

---

### Official Review · Reviewer_Kr8i · 2024-11-03

**Soundness:** 3
**Presentation:** 3
**Contribution:** 3
**Rating:** 6
**Confidence:** 4

**Summary:**

This paper aims to promote fairness in federated learning for medical applications and one FL dataset designed for group fairness is proposed by this paper. In addition, this paper proposes a fairness-aware LoRA and the proposed method shows better performance than compared methods.

**Strengths:**

- Fairness is an important topic.
- The dataset is important for the community.
- The proposed method shows improved performance.
- Both the dataset and code have been released.

**Weaknesses:**

- It is not clear how many sites/institutions are involved in this dataset.
- The annotation and processing protocols should be discussed.
- The attributes-specific singular value is transferred during the communication. This may raise privacy concerns.
- There are six attributes, but only three of them are evaluated in the experiments.

**Questions:**

Please see the weakness part.

---

### Official Review · Reviewer_pqbT · 2024-11-04

**Soundness:** 2
**Presentation:** 2
**Contribution:** 2
**Rating:** 3
**Confidence:** 4

**Summary:**

This paper addresses the fairness issues associated with federated learning in the context of medical images. Specifically, the authors introduce a benchmark dataset aimed at promoting fair federated learning for medical images, comprising paired 2D fundus images and 3D OCT images. Additionally, the authors propose a LoRA-based strategy to mitigate fairness challenges in federated learning settings. Experimental results on this benchmark dataset demonstrate the proposed method in comparison to established baselines.

**Strengths:**

1. The authors focus on fairness in medical image analysis, a crucial area in machine learning for healthcare.

2. The introduction of a benchmark dataset for fair federated learning with eye images is a valuable contribution.

3. The provision of code and data enhances reproducibility.

**Weaknesses:**

1. The rationale for the proposed method lacks clarity. Specifically, it is unclear how decomposing the singular value matrix S into distinct matrices for each demographic group during low-rank adaptation can enhance fairness. The authors should clarify what fairness notions their method can improve and provide theoretical backing for their design choices.

2. While the authors assert that achieving fairness in federated learning with medical images presents unique challenges, they do not specify these challenges. A discussion comparing the complexities of fair federated learning in medical contexts to general settings is absent. Given existing research on fairness in federated learning, such as [1] and [2], it would be beneficial to evaluate the applicability of these studies to medical settings, as the lack of this comparison leaves the motivation for the proposed method ambiguous.

3. The claim that existing state-of-the-art federated learning methods excel with natural images but struggle with medical images is made without a corresponding strategy in the proposed method to address the specific characteristics and challenges of medical data. The authors should clarify how their approach is tailored to meet these unique needs.

4. The authors claimed that in the medical field, cross-institution demographic differences are common due to geographic heterogeneity. However, the cross-site distribution in the proposed dataset, according to Figure 2, doesn't show clear difference among different site. This hinders the utility of the proposed benchmark for site heterogeneity.

5. The creation of the proposed dataset is not clearly described, and many details are missing. Where do these data come from - a single site or multiple sites? What are the criteria to split the dataset into three sites? If all the data involved comes from a single site and is synthesized from three sites, It will be far from the real-world setting as data from different sites can be extremely different as they may use various devices and protocols, and the patient distribution varies a lot.

6. The authors mentioned previous fair federated learning research [1,2]. However, these studies were never included as the baseline. Therefore, there's no insight into how medical applications are different from general settings and whether or not previous methods work for medical applications. Moreover, another trivial baseline missing is using fairness training strategies in clients, such as regularization, resampling, etc.

7. Site fairness is not discussed in the manuscript. However, this could be a problem in the medical field as data disparity between medical sites can be significant. The reason that there's no much performance difference between sites could be that the dataset involved comes from single site.

8. It is counterintuitive that the proposed federated learning method outperforms single-site training configuration. There ought to be more discussion.

9. Standard deviation or statistic testing is not presented to show the significance of performance differences.

References

[1] Ezzeldin, Y. H., Yan, S., He, C., Ferrara, E., & Avestimehr, A. S. (2023, June). Fairfed: Enabling group fairness in federated learning. In Proceedings of the AAAI conference on artificial intelligence (Vol. 37, No. 6, pp. 7494-7502).

[2] Badar, M., Sikdar, S., Nejdl, W., & Fisichella, M. (2024, March). FairTrade: Achieving Pareto-Optimal Trade-Offs between Balanced Accuracy and Fairness in Federated Learning. In Proceedings of the AAAI Conference on Artificial Intelligence (Vol. 38, No. 10, pp. 10962-10970).

**Questions:**

Please see Weaknesses section.

---

### Official Review · Reviewer_H6Db · 2024-11-05

**Soundness:** 2
**Presentation:** 2
**Contribution:** 2
**Rating:** 5
**Confidence:** 2

**Summary:**

The paper, "FairFedMed: Achieving Equity in Medical Federated Learning via FairLoRA," addresses fairness in medical federated learning (FL), particularly concerning demographic diversity. It introduces the FairFedMed dataset, the first FL dataset focused on group fairness in healthcare, featuring 2D SLO and 3D OCT images from 15,165 glaucoma patients across various demographics, including age, race, and ethnicity. The authors propose a new framework, FairLoRA, which leverages low-rank adaptation to customize singular value matrices for each demographic group. This customization preserves demographic-specific features while sharing knowledge across groups to achieve fairer outcomes. Experiments on FairFedMed demonstrate that FairLoRA outperforms traditional FL models and prompt-based models in both classification accuracy and fairness across demographics, proving effective for healthcare applications where demographic equity is crucial.

**Strengths:**

innovatively addressing fairness in medical FL by tailoring demographic-aware adaptations. Its method is robust, with comprehensive experiments affirming the effectiveness of FairLoRA across multiple demographic attributes. Clear explanations and visual aids enhance readability, particularly in illustrating demographic fairness challenges. This work significantly advances health equity by mitigating demographic biases in federated learning, offering valuable contributions to healthcare AI and other sensitive domains.

**Weaknesses:**

The paper’s limitations include FairFedMed’s focus on glaucoma, limited fairness metrics, high computational costs, need for deeper ablation studies, lack of longitudinal fairness analysis, and limited interpretability insights for real-world deployment.

**Questions:**

Testing FairLoRA on other medical datasets, like radiology, would highlight its generalizability beyond ophthalmology. Including additional fairness metrics, such as demographic parity, could provide a fuller view of its fairness. Addressing computational trade-offs and potential optimizations would improve FairLoRA’s practicality, especially in resource-limited settings. More ablation studies on specific components, such as customized versus shared matrices, could clarify each design choice’s impact. A longitudinal fairness analysis across federated rounds, particularly with shifting demographics, would show FairLoRA’s stability over time. Insights on how demographic adjustments impact interpretability would benefit healthcare stakeholders needing transparency. Finally, expanding FairFedMed’s demographic attributes would better represent real-world diversity, enhancing its overall relevance.

**Details Of Ethics Concerns:**

Ethical concerns include potential demographic biases in model predictions, risks to privacy and data security given the sensitive medical data, and the need for transparency in data handling. Releasing the FairFedMed dataset should ensure privacy, ethical consent, and fairness considerations to prevent discrimination or misuse in healthcare applications.

---

### Note · Authors · 2024-11-14

I have read and agree with the venue's withdrawal policy on behalf of myself and my co-authors.